# What Do Childcare Providers Know about Environmental Influences on Children’s Health? Implications for Environmental Health Literacy Efforts

**DOI:** 10.3390/ijerph18105489

**Published:** 2021-05-20

**Authors:** Brenda D. Koester, Stephanie Sloane, Elinor M. Fujimoto, Barbara H. Fiese, Leona Yi-Fan Su

**Affiliations:** 1Family Resiliency Center, Department of Human Development and Family Studies, University of Illinois Urbana-Champaign, Urbana, IL 61801, USA; ssloane2@illinois.edu (S.S.); bhfiese@illinois.edu (B.H.F.); 2Department of Kinesiology and Community Health, University of Illinois Urbana-Champaign, Urbana, IL 61801, USA; efujimo2@illinois.edu; 3Charles H. Sandage Department of Advertising, University of Illinois Urbana-Champaign, Urbana, IL 61801, USA; lyfsu@illinois.edu

**Keywords:** health literacy, environmental literacy, risk communication, children’s health, childcare providers, environmental health literacy, health communication

## Abstract

Children are uniquely vulnerable to toxicant exposures in their environment, which can have long-lasting impacts on their health. Childcare providers are an important population to target for environmental health literacy, as most children in the United States under five years of age spend a significant number of waking hours in non-parental care. There is an increasing body of evidence that children are exposed to toxicants in the childcare environment, and yet little is known about what childcare providers know about environmental influences on the health of children in their care. We conducted semi-structured interviews with 36 home- and center-based Illinois childcare providers to better understand their knowledge, attitudes, and behaviors as they relate to environmental influences on children’s health. We found that the majority of providers had a low level of understanding of potential sources of exposure in the childcare environment, and they did not feel that environmental exposures posed a significant risk to children. Future efforts to increase environmental health literacy should focus on raising awareness and knowledge of environmental health issues for childcare providers before addressing ways that providers can reduce or prevent toxicant exposures to children in their care.

## 1. Introduction

Toxicant chemicals and heavy metals that have been shown to have adverse effects on human health are ubiquitous in the environment [1,2]. Evidence continues to mount that exposure to these toxicants, even in small doses, can have long-lasting impacts on human health, potentially contributing to diseases that are asymptomatic until adulthood [3,4]. Children are uniquely vulnerable to toxicant exposures in their environment [5,6]. The first thousand days of a child’s life, starting with conception, is a period of rapid growth and development of their organs, brains, and metabolic and immune systems [7]. Children eat, drink, and breathe more per kilogram of body weight than an adult, therefore increasing the amount of exposure they have to toxicants in their environment [8]. This combined with developmental behaviors, such as mouthing and crawling, put them at an even greater risk for exposure [6,9,10].

The danger to children from lead, a heavy metal toxicant, has been known for decades. There is significant evidence that lead exposure results in long-lasting neurological damage, and mental and behavioral problems [11]. There is even evidence linking lead exposure in childhood to violent behavior during young adulthood [12]. Children can be exposed to lead from many sources, including dust, soil, deteriorating lead-based paint, consumer products, and imported toys and jewelry [11,13,14]. Lead exposure from drinking water sources continues to be problematic, as recently highlighted by the water lead crisis in Flint, Michigan [3,15]. There are steps that have been taken to reduce exposure, such as the elimination of lead in paint and gasoline. However, half a million children in the United States ages 1 to 5 years have blood lead levels above the reference level of five micrograms per deciliter (µg/dL), as set by the Centers for Disease Control and Prevention [16].

Other toxicants that can pose a danger to health are manmade chemicals, such as phthalates and bisphenol A (BPA). These are found in everyday items, such as personal care products, food and drink, products with fragrance, and children’s toys [17,18]. Exposure has been associated with adverse health outcomes, ranging from developmental delays; learning problems; cancer; respiratory problems, such as asthma; cognitive and behavioral problems; autism; early onset of puberty; and reproductive problems [19].

Multiple toxicants are routinely found in a mixture within one exposure source, such as air pollution. Specific pollutants found in indoor and outdoor air varies between locations but frequently includes toxicants, such as polycyclic aromatic hydrocarbons (PAHs) and volatile organic compounds (VOCs), along with other heavy metals, such as lead [20,21]. Exposure to air pollution has been associated with adverse health outcomes, including respiratory problems, such as asthma; learning deficiencies; cardiovascular disease; and preterm birth outcomes [22,23]. Healthy People 2030 specifically lists air and water pollution as examples of social determinants of health (SDOH) that can contribute to health disparities [24].

The growing body of research on the impacts of exposures on children’s health from lead and other toxicants led to a Federal Executive Order (p. 1, [25]), which called to “make it a high priority to identify and assess environmental health risks and safety risks that may disproportionately affect children”. This, in turn, led to the establishment of the Children’s Environmental Health Research Centers (CEHRC), which were jointly funded by the United States Environmental Protection Agency (EPA) and the National Institutes of Environmental Health Services (NIEHS) to specifically research children’s health outcomes from exposures to environmental toxicants.

Among the objectives of these CEHRC was to engage communities to increase awareness and knowledge of exposures and to educate them on how to reduce exposures [19]. These efforts are part of an emerging concept called environmental health literacy (EHL) that draws from health literacy, risk communication, and environmental sciences [26]. Low levels of health literacy are linked with poor health outcomes [27,28], and increasing health literacy is important in addressing health equity and health disparities [29,30]. Increasing the health literacy of the U.S. population is identified as a high-priority public health issue for Healthy People 2030 [24]. Similar to the field of health literacy [31], there is not one generally accepted definition of EHL [32], but it is thought to encompass a wide range of domains, skills, knowledge, and behaviors related to toxicant exposure and human health [33]. For the purpose of this study, we consider the baseline definition offered by Finn and O’Fallon (2017), “an understanding of the connection between environmental exposures and human health” (p. 496, [33]).

There is a need to improve the EHL of individuals who care for children by providing education on the sources of toxicant exposures, their health outcomes, and steps that can be taken to reduce exposure [33]. Much of the previous work has focused on educating parents and health care professionals. Childcare providers are also important to target for information about protecting children from environmental influences, as the majority of children under the age of five spend a significant portion of their waking hours in non-parental care in a range of care settings, such as day care centers, preschool programs, and family day care homes [34,35].

Socio-ecological models propose that children’s health and well-being is the result of both proximal influences in the family as well as more distal influences, such as neighborhoods and schools [36]. Childcare settings have been specifically identified as important ecologies that can influence children’s health and wellbeing through their effects on children’s nutrition, physical activity, socio-emotional development, and opportunities to educate parents about general development [37,38]. Therefore, it is reasonable to consider whether childcare providers may be a viable target for intervention to influence knowledge and practice in reducing exposure to environmental toxins during the vulnerable preschool years.

Little is known about what childcare providers currently know about environmental influences on children’s health. Current U.S. regulations and policies mostly address physical safety and vary from state to state, as there are no federal childcare licensing standards for environmental health [39,40]. A recent report on state regulations addressing environmental toxicant influences in childcare indicated that most states do not have regulations beyond lead for childcare facilities [41]. However, there is a growing body of literature documenting that children in childcare facilities are exposed to a wide range of toxicants, including polycyclic aromatic hydrocarbons (PAHs), volatile organic compounds (VOCs), perfluorooctane sulfonic acid (PFAS), lead, arsenic, polybrominated diphenyl ethers (PBDEs), and organophosphate (OP) pesticides [18,42,43,44,45,46,47,48].

While there have been some studies of pregnant women [49] and general community members [50,51], we know of only one study with childcare providers that investigated knowledge, attitudes, and behaviors [52], as they relate to environmental influences on children’s health. Additionally, the term “environment” is used extensively in the childcare community to refer to physical structures, such as play equipment, or the learning or socio-emotional environment. We hope to address this gap in the literature.

This study seeks to better understand how childcare providers conceptualize “children’s environment” and the influences in that environment that might impact the health of children in their care. We also sought to understand what childcare providers know about exposures, their effects on health, and steps they can take to reduce exposures. While EHL has been applied at individual and community levels, for this study, we focus on the individual childcare provider.

## 2. Materials and Methods

We used a semi-structured interview format where interviewers ask participants open-ended questions from topics in an interview guide and then follow up with probes. This approach allows respondents to elaborate and discuss additional topics that they find important [53]. Prior research has indicated that qualitative methods are useful in garnering data on attitudes and knowledge about environmental health that can provide background for further research [54] which can inform health literacy and health promotion strategies. The main goal of the interviews was to assess participants’ knowledge about environmental influences on children’s health in more depth than would have been possible with a quantitative method, such as survey research. This approach allowed researchers to capture insight on behavior, perceptions of risk, and other considerations [54]. The University of Illinois Institutional Review Board approved the study.

Interview participants were recruited in two ways. We drew from respondents from a previous survey of center- and home-based childcare providers in Illinois who had agreed to be contacted for an in-depth phone interview. The list of potential respondents was randomized, and a member of the research team contacted individuals sequentially. We also recruited participants through an email that was sent to childcare providers in Illinois via the Illinois Network of Child Care Resource and Referral Agencies (INCCRRA). Of those contacted and scheduled, 36 providers were able to complete the interview, at which point it was determined that we had reached theoretical saturation [55].

A PhD level researcher trained in the Ecocultural Family Interview protocol [56] conducted the phone interviews. The Ecocultural Family Interview protocol is an open-ended, semi-structured conversation that encourages conversational dialogue between the interviewer and the participant. This approach is designed to ensure that participants were comfortable and free to talk about their knowledge, or lack of knowledge, about environmental influences on children’s health.

Interviews lasted about one hour and were scheduled based on the childcare provider’s availability during the fall of 2017. Research staff conducted interviews in a private office using Lync software. The digital audio recorder Audacity^®^ was used to record all interviews. All participants were provided with a USD 25 Amazon e-gift card for participation.

The interviews covered a variety of topics, including initial interest in childcare, general questions about the environment, health questions related to environmental exposures, specific knowledge of chemicals, toxins and sources of exposure, and actions taken to protect the health of children in their care. For this paper, we focus on questions that revealed participants’ knowledge or lack of knowledge about the meaning of children’s environmental health as well as their knowledge about specific sources of exposure.

A professional transcription service transcribed the audio interview files. The interview transcriptions were analyzed by the research team using a semantic approach to thematic analysis. The semantic approach of analysis allows researchers to identify, summarize, and interpret the explicit meanings of the data [57]. Using a qualitative and mixed methods online coding program, Dedoose^®^ [58], the primary coder (who also conducted the interviews) identified major themes in a subset of transcripts and then created initial content categories by identifying and highlighting individual statements in that same subset. A coding structure was developed from the identified categories. The remaining transcripts were coded systematically using the identified coding structure including new codes as they emerged. The research team reviewed the codes and themes to confirm agreement [59], and any discrepancies were discussed until consensus was reached. A second coder (who did not review the codes with the research team) then duplicate coded 25% of the transcripts. Inter-rater reliability was established by the primary and secondary coders through reviewing the codes together and discussing any discrepancies until they reached final consensus.

## 3. Results

### 3.1. Childcare Provider Characteristics

Demographic characteristics are presented in Table 1. Participants were experienced in providing childcare with half (50%) having worked for more than twenty years as a provider. Twenty-eight percent of participants had a degree in childcare and worked at a center-based day care (10/36), and 17% of participants had a degree in childcare and worked at a home-based day care (6/36). Fifty-eight percent had some form of childcare professional accreditation, such as NAEYC [60] or Excelerate [61]. Providers frequently mentioned that they had always been interested in providing childcare (53%) and that they continued to care for children because they had a passion for it (56%). We did not collect demographic information about provider age, race, or ethnicity.

### 3.2. Meanings of Environment

At the beginning of the interview, participants were asked “What does the word ‘environment’ mean to you?” The question was intended to be broad, but participant responses fit into one or more of three categories: physical environment, social–emotional environment, or intellectual–learning environment. The majority of the respondents (92%) covered aspects of the physical environment, such as children’s surroundings; the spaces they use during the day; where they nap, play, and eat; the outdoors; the playground; and the community where they resided. Over half (58%) gave responses that related to the social and emotional environment, including the presence of loving teachers and friends, children feeling emotionally safe, and having close relationships with teachers and friends. Responses that addressed the intellectual and learning environment (28%) focused on opportunities for learning, exploration, and growth, such as educational materials.

### 3.3. Meanings of Children’s Environmental Health

Participants were asked next about the term “children’s environmental health”. Close to half (42%) said they had never heard the term or were unsure if they had. All participants were further probed what they thought the term children’s environmental health meant (e.g., “What does that team mean to you?” “What do you think of?”). Fifty-three percent of participants talked about the physical environment. They specifically mentioned children’s safety (e.g., keeping them free from harm and sickness/acute sickness), keeping the house clean, getting outside enough, recycling, and poison control.

“To me [children’s environmental health] would mean that the environment they are in protects them, keeps them safe, whether from germs or just regular harm and accidents”.(Participant #171)

More than one third of participants (39%) talked specifically about the physical health of children: getting well checks at the doctor, shots, germs, adequate nutrition, and exercise. Social-, emotional-, and mental health-related responses to what comprised children’s environmental health were also given by 39% of participants: positive interactions between teachers and students and among students, conscious discipline, mental health of parents, and the home environment (e.g., abuse and neglect).

“So environmental health would be like being able to interact with other kids in your environment. For the preschool setting, it would be the kids being able to do their puzzle and ask for help from one another and/or teachers. And it being healthy, not negative, but always positive, encouraging. And that includes the kids and the way they speak to one another...”(Participant #102)

Only 25% of participants mentioned air quality, water quality, products, and chemicals. These participants talked about radon testing, smoke, dust, lead paint, cleaning products, food packaging, chemicals in rugs and furniture, and water quality.

“To me, that means that you need to keep a healthy environment, like no lead paint, cleaning up the house-I mean, you don’t have to keep it spic-and-span, but you wanna keep it clean where the carpets are clean, or the hardwood floors are clean. So they’re not sharing dirt. I have a couple of kids with allergies, so I keep the dust swept up weekly.”(Participant #180)

### 3.4. Impact of the Environment on the Health of Children

Participants were also asked “Do you feel that the environment has any impact of the health of the children in your care?” At this point in the interview, participants had been encouraged to think about environmental health hazards, rather than the social or emotional environment. The most frequent response, given by 33% of participants, was that air quality had a negative impact on health. Participants’ concerns included smoking, perfume, radon, outdoor air quality (connected to asthma), chemicals from crops, and seasonal allergies.

Concern about chemicals was the next most frequent response, given by 28% of participants. These concerns included carpet cleaner, perfume, strong cleaners, air fresheners, bleach, lead, red food dye, and pesticides. However, almost as many participants were not concerned about the environment’s impact on the health of the children in their care.

“I don’t really have any concerns as far as that goes [air quality, water quality, products, chemicals] because I don’t worry about that kind of stuff because you can’t do anything about it anyway. So, I just feel like if I can’t do anything about it, why worry about it? There are too many other things to worry about in this day and age.”(Participant #175)

A few (19%) participants were concerned about water quality, citing poor city water, contaminated well water, and old pipes as the basis of their concerns, but a higher amount (28%) specifically said they were not concerned about water quality.

“The water is city water, so I have to trust those people are following the guidelines, so I can’t really worry about that.”(Participant #102)

Providers were then read a list of toxicants and potential sources of toxicant exposure and asked whether they considered the toxicant or source as a problem in the spaces where they cared for children (not in the world at large) and to what degree. Participants were asked to indicate whether they felt the toxicant or source was a very serious problem, a moderate problem, not a problem, or if they did not know. A complete list of exposure sources and toxicants that participants were asked about are listed in Table 2. In some instances, providers mentioned that an exposure or toxicant was not a problem because they felt they were taking actions to address the problem. In some instances, the steps they indicated taking would likely protect children from exposures, whereas with others, it was less clear.

“We actually are actually a scent free facility. So, we don’t use any sort of scents or lotions, or anything like that.”(Participant #137)

“Well everything says non-toxic, so I’m assuming they’re [art supplies] good.”(Participant #165)

## 4. Discussion

To our knowledge, this is the first study to conduct in-depth interviews with childcare providers to assess their knowledge of children’s environmental health as well as their level of concern about potential risks posed to children by environmental exposures. We interviewed 36 center- and home-based childcare providers. Participants were mostly unconcerned with environmental exposures and did not view them as posing a significant risk to the children in their care. Only 25% of providers spontaneously mentioned environmental toxicants (e.g., water quality, chemicals, and air pollution) as things that they associated with environmental health. Participants were more concerned with physical safety related to accidents, or acute injury (e.g., making sure electrical outlets are covered and protecting against falls) and with social and emotional development. Close to half of the participants had never heard the term children’s environmental health or were unsure of its meaning.

Researchers and advocates frequently use the term “children’s environmental health” in their outreach and education materials. Among the hallmark features of the previously mentioned Children’s Environmental Health Research Centers is their Community Outreach and Translation Cores. These cores are charged with identifying a particular target community of focus. They are then charged with taking research findings, translating them into a “language” that their target community understands, and then engaging with the community. Other groups providing education and information include the Children’s Environmental Health Network (CEHN), a private 501(c)3 organization that is focused on protecting children’s health from environmental factors, and the Pediatric Environmental Health Specialty Units (PEHSUs), a network of trained environmental health physicians available to answer questions from parents, public health officials, and the general public.

It is notable that all three of these entities have the term “children’s environmental health” in their name. Even more crucially, most of the messages and outreach materials provided by these groups use the phrase “children’s environmental health”. Our research suggests that how researchers and practitioners define “environment” may be at odds with definitions used by their target audience. This emphasizes the need for environmental health researchers and practitioners to explicitly define what they refer to as the “environment” (e.g., toxic chemicals and heavy metals).

The confusion around terminology and childcare providers’ perceptions of low levels of risks posed by toxic chemicals has serious implications for risk communication. In order to understand the threat and susceptibility to a toxicant exposure in their environment, a childcare provider would need to have a basic grasp of the components of exposure science. The basic components entail the following: (1) there are chemicals present in their environment; (2) there are pathways to exposure (e.g., dermal, aspiration and oral); (3) exposure to these chemicals, even in low doses, can impact health; and (4) the health effects of these exposures may not be evident until later in life [51].

Gray (2018) proposed a model of EHL that features awareness and understanding of how environmental exposures influence health (such as the basic components listed above) as the foundation of EHL [32]. From there, individuals can learn skills and self-efficacy to make choices to protect their health, then go further to participate in community change or collective action to reduce exposures on a larger scale. A pilot intervention with childcare providers showed that increasing knowledge about environmental health hazards led to increased protective behaviors [62].

Our research provides support for the need to raise childcare providers’ awareness and knowledge of environmental influences on children’s health before starting efforts to build skills and self-efficacy. Clearly defining the term environment in environmental health communication is critical. These efforts can lead to greater EHL and ultimately greater protection from environmental exposures to children in childcare.

## 5. Conclusions

In conclusion, this study sought to characterize how childcare providers conceptualized the environment in the context of environmental influences on children’s health. We found that the majority of childcare providers had a low level of understanding about toxic exposures in the childcare environment, and they did not consider these exposures to be a threat to the health of children in their care.

A limitation of this study is that we focused only on individual childcare providers’ knowledge and understanding of environmental influences on children’s health and did not address the larger context of childcare operations, regulations, policies, and procedures. These factors and their contribution to reducing toxicant exposures in the childcare setting are all important to consider [39] but are outside of the scope of this paper. Future research should consider these factors. An additional limitation is that we collected minimal demographic information from participants. It will be important for future research to examine if there are differences in EHL among providers given their age, race, ethnicity, and geographical location (e.g., urban and rural). These differences may identify factors contributing to health inequities.

Gray and Lindsay (2019) call for further research to understand how individuals and communities move along the EHL continuum [63]. In order to adequately assess this movement, or gain awareness and knowledge, we will need to be able to measure baseline EHL. While there have been some nascent efforts to develop a validated environmental health literacy measure [51,64,65], we encourage researchers to continue these efforts and to address validity and reliability with additional populations. This measurement challenge is faced in health literacy, and the lack of an accepted valid and reliable tool is a limitation for the field [27].

Communication research, particularly health communication, is uniquely positioned to inform both the theories and methods of researchers studying (1) the public understanding and perceptions of environmental health; (2) how such literacy and beliefs are formed; and (3) strategies to boost them [66,67,68]. Effective communication about risks and protective actions are critical to increasing EHL. An initial step should be the establishment of a plain language definition of environmental health literacy.

Among the goals of this study was to inform the development of environmental health communication messages targeting childcare providers. Messages should be tested for acceptability, knowledge change, and intent to change behavior. They should also be developed using plain language guidelines [69], which can be an effective way to increase health literacy [70,71].

In summary, future directions for this work should entail establishing a plain language definition of environmental health literacy, developing a valid and reliable measure of environmental health literacy, developing messaging materials, and identifying trusted sources of and channels for information.

## Figures and Tables

**Table 1 ijerph-18-05489-t001:** Participant demographic characteristics.

	*n* (%)
Female	36 (100%)
Length of time as childcare provider	
0 to 5 years	3 (8%)
6 to 10 years	5 (14%)
11 to 15 years	5 (14%)
16 to 20 years	5 (14%)
More than 20 years	18 (50%)
Childcare setting	
Childcare center	11 (31%)
Church or faith-based center	5 (14%)
Home-based	20 (56%)

**Table 2 ijerph-18-05489-t002:** Degree of provider concern about toxicants and sources of toxicant exposures in their own childcare environment.

Source	Not a Problem	Moderate Problem	Very Serious Problem	Do not Know
Lead (paint)	97%			3%
Asbestos	92%			8%
Contaminates in drinking water	94%	3%		3%
Mold (inside)	86%	8%	3%	3%
Air pollution inside	83%	14%		3%
Pesticides	83%	17%		
Chemicals in art supplies	78%	16%	3%	3%
Chemicals in rugs and furniture	75%	11%		14%
BPA	75%	11%	3%	11%
PBDEs	73%	9%		18%
PFCs	69%	17%	3%	11%
Chemicals in personal care products	67%	25%		8%
VOCs	68%	9%		23%
PCBs	58%	20%		22%
Chemicals in children’s toys	56%	19%	3%	22%
Pesticides, hormones, antibiotics in food	55%	31%	3%	11%
Air pollution outside	50%	47%		3%
Dust	50%	42%	8%	
Triclosan	57%	20%		23%
Phthalates	46%	20%		34%

Notes: polychlorinated biphenyls (PCBs); perfluorinated chemicals (PFCs).

## Data Availability

The interview guide and numeric data are available upon reasonable request to the authors. The interview data are protected.

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
