# Peer review of "What Do Childcare Providers Know about Environmental Influences on Children’s Health? Implications for Environmental Health Literacy Efforts"

_ijerph, 2021, doi:10.3390/ijerph18105489_

Round 1
Reviewer 1 Report
Authors have provided adequate responses to reviewer questions and comments.
Author Response
Reviewer 1 - Thank you for your feedback during the first round of reviews which helped to strengthen our manuscript. We hope that it will make a strong contribution to the literature on environmental health literacy and equity.
Reviewer 2 Report
This revision uses much clearer language and it is clear that my previous comments have been addressed. I am happy with the revisions, and I have no concerns.
Author Response
Reviewer 2 - Thank you for your suggestions and comments in the first round of reviews which helped to strengthen our manuscript. We hope that this paper will make a strong contribution to the literature on environmental health literacy and equity.
Reviewer 3 Report
The authors have addressed most reviewer concerns. It is easier to understand the larger context in which this work is situated as well as the context of this particular area of inquiry. The text needs another copy edit which I'm sure will happen in the manuscript preparation stages with the journal (words missing, incorrect usage, punctuation). There are a few remaining issues which I would like to see addressed:
- since demographics (e.g. age, race, geography [urban, rural, suburban], etc) were not collected - this needs to be transparent and noted, perhaps in the section introducing Table I and/or in the limitations, including future directions. Again, directly addressing health equity.
- this point - which was clarified in the response but not in the paper. Needs to be addressed, discussion seems most appropriate place. It strengthens the argument for the need for this work. See copy of issue/response here.
Are these themes from everyone? Or only (line 205) the 25% of participants who mentioned air quality, water quality, products and chemicals? The impact themes seem quite focused on environmental health when only 25% of participants overall defined “environmental health” as air quality, etc. Please explain.
RESPONSE: Thank you for requesting this clarification. The goal of our study was to assess how much child care providers know about environmental health. As the interview progressed, the questions became more and more specific regarding environmental toxicants and exposures. Therefore, the themes are from all participants, despite the fact that, initially, only 25% spontaneously mentioned air quality, water quality, products and chemicals.
- Table II is improved. However, it is still unclear in some aspects and requires further clarification.
For example, percentages across rows don’t add up to 100% - were there nonresponses? Please indicate N or give other clarification.
Author Response
Please see the attached PDF file.

This manuscript is a resubmission of an earlier submission. The following is a list of the peer review reports and author responses from that submission.
Round 1
Reviewer 1 Report
This is a very well conceived and methodologically meticulous study. I am interested to know what was the response rate to get to N=36. Authors may want to comment on selection bias/ self-selection bias. Somer readers may want to look at the semi-structured interview questions/ form used by the authors. Were all interviews conducted by the same, single interviewer?
Author Response
Thank you for your helpful comments. Please see the attachment for our point-by-point response.

Reviewer 2 Report
The manuscript provides support to increase awareness among childcare providers, as well as it gathers basic knowledge of environmental influences on children’s health in order to develop skills and self‐efficacy to reduce toxic substances exposures.
Introduction provides a good explanation of the problem and the other sections are also very clear, however I recommend that in ‘Materials and Methods’ more information about the sample and characteristics of the responders’ participants should be added for a more completed and clear understanding. The only information about the sample to know is ‘36 providers were able to complete the interview’ (line 128) and nothing else. Why is the sample size 36 and not 50? Does the study describe the eligibility criteria for participant selection? What are participants’ characteristics?
As long and it is provided an adequate extended description of the sample characteristics, this manuscript can be acceptable for publication.
Author Response
Thank you for your helpful comments. Please see the attachment for a point-by-point response.

Reviewer 3 Report
This manuscript talks about the extremely important topic of "environmental health literacy". It fills a gap about the knowledge that child care providers have about the meaning and impact of children's environments on their health.
Nevertheless, I found the manuscript to lack clarity in many areas, including framing the topic, situating it within the relevant health literacy and health equity fields, explaining the methods and results, and interpreting the results in light of the field and next relevant steps.
Here are detailed comments:
Introduction:
- Extensive and clear overview of the relationship between environmental toxicants and children’s health
- Line 79: need to improve environmental health literacy of individuals who care for children by providing education
- Most of previous work: focused on parents and health care professionals
- Line 88 - Gap: what do child care providers know about environmental influences on children’s health
- Line 91: Most states do not have regulations beyond lead, for child care facilities
- Line 95: please define acronyms at first use – and then use uniformly throughout (e.g. PFAS, PBD’s, PBDEs)
- Line 107-109: emerging used in same sentence twice, please choose a different word for one of the occurrences
- Final paragraph of the introduction: After the clear establishment of the relationship between environmental toxicant levels and children’s health, a framing of environmental health literacy within the context of health literacy should be included. As I understand it, this is a special issue focused on health literacy and health equity. It is not clear from the manuscript how this fits into that larger special issue focus.
Notes on this: As it is currently discussed in the final paragraph of the introduction, environmental health literacy is essentially the knowledge of child care providers – and wanting to understand what they know in order to educate them and therefore increase their understanding of the issue. Health literacy is a multidimensional concept (see new Healthy People 2030 definitions and related briefs, for example) that incorporates knowledge and skills for tackling any health issue and the decisions/actions that surround it as well as many other components across and within levels. Please specify, what is the definition of health literacy being used, in the context of the health literacy field? If this is “environmental health literacy” as a concept around individual knowledge alone, please call it that within the context. If not, please specify more fully what this emerging concept of environmental health literacy is – or could be - and how it fits into a health literacy framework, per Finn & O’Fallon or others. For example, what are the “levels” operating here? In a social ecological frame, we understand the child nested within the family nested within school/community environments, nested within institutions, nested within state and federal environments/policy. Please locate this work. Is it a focus on what child care professionals know within the school/community environments – and/or is it about the actionability and impact of this “knowledge”, or both? What literature supports that focusing on the knowledge of these professionals will impact outcomes for children related to toxicants and health, or similar if there is no specific literature on this yet (for example, you talk about work in this area with parents and health care professionals)? What is the result of “educating” these child care providers? What theory and frameworks (what are they?) guide the research question? Certain theories and frameworks would guide a review of institutional level health literacy practices and guidelines, which might include educating professionals. But, such health literacy theory and frameworks might also specify the working environment and responsibility of institutions in providing healthy work and care environments, which is beyond the knowledge of providers alone. Perhaps, for example, it is the processes, procedures, and physical design of child care spaces that impact children’s health as related to toxicants? Please make it clear for the reader why understanding “how child care providers conceptualize “children’s environment” and the influences in that environment that might impact the health of children in their care” is related to the relationship between toxicants and children’s health. Some orientation is needed in the introduction which can be followed through on in framing the discussion – using what the research can teach us.
Materials & Methods:
- Line 112: what does “deeper level” mean?
- Line 112-115: Please clarify importance related to it being an emerging field – Is it an emerging field or an emerging concept? What about it being emerging necessitates this work? The second half of the sentence does not follow from the first part.
- Utilized = use; plain language please
- 121: are these health literacy and promotion strategies or health and promotion strategies?
- Line 139-140: gift card for participating (i.e. paid to participate) or gift card as a thank you for time/acknowledgement of effort [typically IRB approves the latter, please clarify]
- Line 144-147: Is this a methodologically driven decision? How done?
- Line 156-159: Please specify. How many people coded each transcript? How were final codes review and how was agreement confirmed? How was inter-rater reliability established and what was it?
Results
- Line 162-163: Please indicate where there is overlap in the 50% and 58% groups?
- Line 167-168: Please indicate the overlap between child care degree and where providers work.
- 1: I’d like to see a demographic table of child care provider characteristics by relevant domains, including age, race, gender, and others, including the ones you’ve mentioned
- 2: Meanings of Environment – there is clearly overlap here – can you discuss that so we can get a sense of how many people had multi-layered definitions, etc for the three types of environment reported.
- Line 170: What was the actual question? Was it the definition of environment in the child care environment context?
- 3 Meanings of Children’s Environmental Health
- Line 184: When further probed what they thought… how were they probed? What was the follow-up?
- Line 186: Is “keeping the house clean” or others related to general children’s environmental health – or was the child care environment specified? For this example, is keeping the house clean about keeping the child’s home clean or is about the house that serves as the child care, as in a family home day care?
- 4 Impact of Environment on Health of Children
- What “environment” is being referred to? Their original definition(s) of environment – the definition they arrived at after probing in 3.3.
- Are these themes from everyone? Or only (line 205) the 25% of participants who mentioned air quality, water quality, products and chemicals? The impact themes seem quite focused on environmental health when only 25% of participants overall defined “environmental health” as air quality, etc. Please explain.
- Please provide the general question asked and answer choices.
- Table I: Source is the first column. Some of these rows seem like sources of toxicants but some seem like toxicants. Please clarify or divide Table I into sources and toxicants.
- Table I: To whom does “not a problem because actions taken” refer to?
- Table I: Very serious problem has additional numbers next to the percentages. I assume these number refer to participant IDs. Please clarify and explain what they are and why they’re there.
- Table I: What do the percentages mean? It can’t be the percent of participants who said lead (paint) were overall not a problem, because the row of responses adds up to more than 100%. What were the response choices? How many people responded to each choice? Were some collapsed?
- For each question described in this section, please provide the full question asked and any probes.
- Please move Line 240-242 above Table I to clarify the data presented. This clarifies a few things noted, but the Table is still unclear overall. You should be able to “read” the Table without much outside information. Please include the needed labels and details in the Table or Table footnotes.
Discussion
- Line 258-265: What do we know about the understandability, accessibility, actionability, etc of these materials or websites? What do assessments say about the “language” they use? i.e. are health literacy principles used to express risk, plan language, understandability, actionability, etc?
- Line 266-272: Are public definitions (per your child care provider interviewees) using “incorrect” environment definitions – or is it that the children’s environmental health term and its use in the field and in practice unclear?
- Line 273: What does low levels of risk refer to? Depending on child care providers definition of children’s environment or children’s environmental health – and which environment they’re thinking of – their assessment of risk might be different. Are low levels of risk referring to child care providers’ assessments of toxicants in Table I?
- How does the child care environment facilitate a good environment for children to create positive outcomes?
- Line 273-281: Why is the locus on the provider alone? What about the operations, processes, and policies in which child care workers provide care that should be reducing risk and exposure for kids and staff? Do staff have the efficacy and choice to make such changes, even if they did understand basic exposure science?
- Line 282-289: I’m not clear that the argument has been made about what this research is telling us. Please clarify. What are next steps?
- Line 296-301: say more. Research, practice, measurement, evaluation? Pilot testing? Desired outcomes of messaging? Assessment of messaging?
- Line 307: What about developing a plain language definition of environmental health literacy?
Overall, please review for clarity of language throughout the manuscript.
Author Response
Thank you for your helpful comments and suggestions for improvement to the manuscript. Please see the attachment for a point-by-point response.
